# Thermal and Mechanical Analysis of Polyethylene Homo-Composites Processed by Rotational Molding

**DOI:** 10.3390/polym11030528

**Published:** 2019-03-20

**Authors:** Antonio Greco, Francesca Ferrari, Maria Grazia Buccoliero, Greta Trono

**Affiliations:** Department of Engineering for Innovation—University of Salento, Via per Monteroni, 73100 Lecce, Italy; francesca.ferrari@unisalento.it (F.F.); mariagrazia.buccoliero@studenti.unisalento.it (M.G.B.); greta.trono@studenti.unisalento.it (G.T.)

**Keywords:** rotational molding, homo-composite, crystalline structure, mechanical properties

## Abstract

This work is aimed at studying the suitability of ultra-high molecular weight polyethylene (UHMWPE) fibers for the production of polyethylene homo-composites processed by rotational molding. Initially pre-impregnated bars were produced by co-extrusion and compression molding of UHMWPE fibers and linear low-density polyethylene (LLDPE). A preliminary screening of different processing routes for the production of homo-composite reinforcing bars was performed, highlighting the relevance of fiber impregnation and crystalline structure on the mechanical properties. A combination of co-extrusion and compression molding was found to optimize the mechanical properties of the reinforcing bars, which were incorporated in the LLDPE matrix during a standard rotational molding process. Apart from fiber placement and an increase in processing time, processing of homo-composites did not require any modification of the existing production procedures. Plate bending tests performed on rotational molded homo-composites showed a modulus increase to a value three times higher than that of neat LLDPE. This increase was obtained by the addition of 4% of UHWMPE fibers and a negligible increase of the weight of the component. Dart impact tests also showed an increased toughness compared to neat LLPDE.

## 1. Introduction

Rotational molding is a pressure-free process, which allows the production of hollow plastic containers with different dimensions. Samples obtained with this technology are free of residual stress and have a uniform thickness. The choice of the material for rotational molding is strictly limited, since the process requires low viscosities, a good sintering of the powders, and the absence of voids. Furthermore, the brittleness of the polymer must be taken into account, in order to avoid rupture after extraction.

Currently, the greatest part of the production by rotational molding involves the use of linear low-density polyethylene (LLDPE); the main drawback lies in the poor mechanical properties of LLDPE, which has a low elastic modulus, and an operating temperature range of 80 to 120 °C.

Several additives, such as inorganic fillers, can be added to improve the mechanical response of the material. On the other hand, the addition of inorganic fillers involves an increase of viscosity, which in turn does not allow for a good distribution of the fillers inside the mold during the process. In addition, the material must be tough enough to allow extraction from the molds [1].

Recently, other approaches were evaluated in order to increase the stiffness of rotomolded containers, which involves the addition of nanofillers [2,3], or reinforcements in the form of particulate [4,5] or short fibers [6,7]. Nevertheless, despite the improvement in stiffness, the incorporation of these reinforcements involves several problems in terms of processability, embrittlement, and voids [8,9]. Moreover, segregation phenomena can arise if powders with a different weight and size are used, with an accumulation of heavier particles on the outside surface of the mold, and the lighter particles accumulating on the internal surface [1,4,10].

Alternatively, reinforced rotomolded products can be produced by the inclusion of long fibers, by using thermoplastic prepreg in a bladder molding process [11,12]. This technology, however, has serious limitations for the production of complex geometries. Additionally, the reinforcement is distributed in the whole product, even where it is not strictly required for the load conditions. This drawback can be overcome by the selective placement of the reinforcement (e.g., glass fibers) distributed only on the zones subjected to high loads, using pultruded rods [13]. This application allows for a reduction in the cost and weight of the final product. Nevertheless, the use of glass fibers creates an increase in mechanical properties only if good adhesion between the matrix and the fibers is achieved. Likewise, the presence of glass fibers may cause distortions or residual stresses inside the component, as well as some increase in the weight.

On the other hand, single-polymer composites (SPCs), or homo-composites, represent an emerging family within the polymeric composite material. Considerable research efforts have been undertaken to produce lightweight, easily reprocessable homo-composites, characterized by matrix and fibers belonging to the same polymer type [14,15]. In particular, polyethylene homo-composites, consisting of ultra-high molecular weight polyethylene (UHMWPE) fibers and different grades of polyethylene (PE) matrix showed very good fiber–matrix adhesion without the use of any chemical treatment. This allowed researchers to obtain homo-composites characterized by excellent stiffness, tensile strength, and impact strength [16], which makes them very attractive for the rotational molding process.

Therefore, this work is aimed towards the rotational molding of LLDPE prototypes, reinforced with UHMWPE fibers, characterized by increased stiffness and toughness, compared to neat LLDPE. Preliminarily, pre-impregnated bars, produced by co-extrusion and compression molding of UHMWPE fibers and LLDPE, were introduced to a rotational molding cycle. An analysis on the quality of the homo-composite bars processed by co-extrusion and compression molding is presented. Subsequently, the reinforcing homo-composite bars were placed inside a cubic-shaped mold and included into the LLDPE matrix during the rotational molding cycle. Homo-composites processed by rotational molding are characterized in terms of bending stiffness and impact resistance, showing the potential of the developed approach for the production of hollow components characterized by high mechanical properties.

## 2. Materials and Methods

In this work, LLDPE, Clearflex RM 50 by Polimeri Europa (Versalis, San Donato Milanese, Italy), was used as a matrix. RM 50 has a melt flow index of 4.2 (ASTM D1238), a density of 0.93 g/cm^3^ (ASTM D 1505) and a melting point of 126 °C. UHMWPE fibers, Endumax TA23 tapes by Teijin (Arnhem, The Netherlands) (55 µm × 2 mm cross section), density 0.95 g/cm^3^, were used as reinforcement.

### 2.1. Processing of Polyethylene Homo-Composites

Rotational molding is a pressure free process in which the only force acting on the material during the whole cycle is the gravitational force. However, in order to produce a polyethylene homo-composite, pressure is required in order to allow molten LLDPE to flow inside the UHMWPE tape and obtain an efficient fiber impregnation. As a consequence, before conduction the rotational molding processing, impregnation of UHMWPE fibers by LLDPE matrix was attained by means of co-extrusion and compression molding.

Due to the quite low melting temperature difference between LLDPE and UHMWPE, particular care must be taken when choosing the appropriate temperatures during the processing of polyethylene homo-composites. Processing of LLDPE, reinforced with inorganic fibers, is usually performed at relatively high temperatures (180 to 190 °C), which allows for a viscosity reduction of the matrix, and therefore for a more efficient impregnation of the fibers. However, processing of homo-composites must be performed at much lower temperatures, in order to preserve the UHMWPE crystalline structure, which enables the retention of the high stiffness of the fibers [14]. Therefore, the temperatures used for homo-composites processing are lower than those used for standard processing of LLDPE. In every processing method described hereafter, such temperatures have been chosen based on preliminary optimization trials, which allowed for a minimization of the changes in the crystalline structure of UHMWPE, yet still retaining a sufficiently low viscosity of the LLDPE matrix.

Two methods were used for the production of polyethylene homo-composite reinforcing bars:The first method involved the production of homo-composite bars with 20% wt of UHMWPE and 80% wt of LLDPE through compression molding, using a Campana hot press. A controlled amount of LLDPE and UHWMPE fibers were placed in a 3 mm thick steel frame, after which samples were compression molded at 135 °C, applying a first pressure step of 50 bar for 5 min and a second step of 100 bar for 5 min. Following this, samples were cooled down to room temperature by means of a hydraulic cooling system, under 100 bar pressure. The thickness of compression molded samples was 3 mm. The choice of the temperature during compression molding was based on a preliminary optimization cycle, showing that this is the minimum temperature required to attain good fiber impregnation. The compression molded samples were labeled as HC_CM.The second method involved the preliminary production of homo-composites through co-extrusion of LLDPE with UHMWPE fibers. Co-extrusion was performed in a Haake R Rheomex PTW16/25 D twin screw extruder. The extrusion process was run at a screw temperature profile of 140–160–170–160–150–150 °C–130 °C with a screw speed of 7 rpm. The extruder was provided with a 3-mm rod die, modified in order to allow for co-extrusion, as reported in the scheme of Figure 1. Essentially, UHMWPE fibers were fed in the extruder chamber though the pressure gauge gate. A co-extrusion element was placed inside the extrusion die throat, which had a diameter of 6 mm. This co-extrusion element was a cylinder, hollow throughout its length, apart from a solid base. On the solid base, which faces the rear of the extruder, a 2 mm diameter hole allowed for fiber inlet to the die. The hollow cylinder was further provided with holes on its side surface, which allowed for molten matrix inlet to the die. In the hollow length of the co-extrusion element, the molten matrix surrounded the fibers, and the coextruded homo-composite was finally passed though the extrusion die. LLDPE matrix, in powder form, was filled in the extruder and melted before the addition of the fibers, which was attained in the modified die. Therefore, it was expected that the fibers would reach a maximum temperature of 130 °C, the die temperature, during their processing. This procedure allowed for us to obtain a homo-composite with a higher amount of UHMWPE fibers, 30%, and 70% LLDPE matrix. Due to the poor impregnation of the UHMWPE fibers after co-extrusion, further compression molding was carried out with the same processing conditions used for HC_CM. However, in this case, the compression molding temperature was set to 125 °C, and the thickness of reinforcing bars was set to 0.4 mm by the use of a thinner steel frame. Co-extruded samples were labeled as HC_CE, whereas co-extruded and further compression molded samples were labeled as HC_CE_CM.

Rotomolded prototypes were produced by using a two axes lab scale rotational molding machine designed and produced by Salentec srl (Lecce, Italy). Samples were obtained by using a box-shape mold with an edge length of 148 mm. The reinforcing homo-composite bars produced by compression molding and co-extrusion/compression molding were bonded at the inner surface of the aluminum mold by means of a silicon adhesive. Following this, 400 g of LLDPE powder was added, which corresponded to a nominal thickness of 3.5 mm of rotational molding prototypes. The oven temperature was set at 140 °C, with a rotation speed of the primary and the secondary axes at 8.2 rpm and 29.8 rpm, respectively. After a heating cycle of 90 min, the mold was cooled for 30 min by forced convection. During the heating cycle, melting and sintering of LLDPE matrix powders allowed for the inclusion of the reinforcing bars in the rotational molded prototype. Compared to a standard rotational molding cycle, which is usually run with much higher oven temperature, a reduction of the oven temperature is required in order to preserve the fibrous structure of UHMPE [17]. Consequently, the residence time in the oven was increased in order to attain complete sintering of the LLDPE powders [18]. The maximum temperature of the inner air during the rotational molding process was measured to be 135 °C.

Different prototypes were produced with different reinforcing layouts and weight percentages of UHMWPE fibers, as reported in Table 1. In all instances, the amount of fibers was very low, which allowed to neglect the weight increase due to the addition of fibers. This also in view of the very similar densities of matrix and reinforcement. On the other hand, the UHMWPE fiber content was higher than that of pultruded glass fibers, as reported in a previous work [13]. In fact, for glass fiber, for amounts higher than 0.6%, the rotational molded prototypes underwent significant distortions, due to the difference in the thermal expansion coefficient between glass and LLDPE. In the case of UHMWPE fibers, no significant distortion of the rotational molded prototypes was observed, as highlighted by the picture of Figure 2. Comparing the different rotational molded prototypes, the fiber content of samples obtained with HC_CE_CM bars was lower than that of HC_CM bars, due to the fact that in the former case 0.4 mm thick bars were used, whereas in the latter case the thickness was 3 mm.

### 2.2. Mechanical and Thermal Characterization

Differential scanning calorimetry (DSC) analysis was performed on DSC1 Star System by Mettler-Toledo (Mettler Toledo, Greifensee, Switzerland) instrument under a nitrogen flux of 60 mL min^−1^, applying a first heating scan between 20 and 210 °C at 10 °C min^−1^, followed by a cooling scan from 210 to 20 °C at 10 °C min^−1^ and a second heating scan up to 210 °C at 10 °C min^−1^.

Tensile properties were carried out on co-extruded and compression molded homo-composites, using a LLOYD LR 5K dynamometer with a crosshead speed of 5 mm/min.

Plate bending tests were carried out on the faces extracted from the rotomolded prototypes. Sheets 100 mm × 100 mm, which can be approximated as thin plates, were simply supported on their perimeter, and loaded with a square punch (16 mm edge) in their center. In reinforced samples, the reinforcing bars were placed at the extrados. The test speed was set to 0.5 mm/min. Due to the high edge to thickness ratio, every sheet can be approximated as thin plates. Therefore, the equivalent elastic bending modulus E_B_ was calculated according to Equation (1) [19]:(1)$$E_{B}=\frac{48\left(1-\nu^{2}\right)}{\pi^{6}s^{3}}\frac{p}{\zeta }\left[\overset{9}{\underset{m=1}{\sum }}\overset{9}{\underset{n=1}{\sum }}\frac{\sin\frac{m\pi x}{a}\sin\frac{n\pi y}{b}}{mn\left(\frac{m^{2}}{a^{2}}+\frac{n^{2}}{b^{2}}\right)}\left(\cos\frac{m\pi \xi_{1}}{a}-\cos\frac{m\pi \xi_{2}}{a}\right)\left(\cos\frac{m\pi \eta_{1}}{b}-\cos\frac{m\pi \eta_{2}}{b}\right)\right]$$
where *p* is the applied pressure, obtained by dividing the force by the punch area, s is the plate thickness, *a*, *b*, $\xi_{1}$, $\xi_{2}$, $\eta_{1}$, $\eta_{2}$ are the geometric properties of the supported plate and the punch, as defined in Figure 3, and ζ is the measured displacement, evaluated at x = *a*/2 and y = *b*/2.

Impact tests were carried out with the Fractovis Plus impact machine on 8 bars-reinforced and unreinforced sheets extracted from rotomolded samples. Two different impact energies (30 J and 40 J) were used for the test.

## 3. Results and Discussion

### 3.1. Thermal Characterization of Polyethylene Homo-Composites

DSC curves of the LLDPE matrix and neat UHMWPE fibers during a first heating scan are reported in Figure 4. As shown, the two polymers, though being composed by the same repeating unit, had significantly different melting behavior. In particular, LLDPE was characterized by a much lower melting temperature, with a peak at 124 °C and a melting enthalpy of 108 J/g, compared to UHMWPE, which was characterized by a melting temperature peak of 152 °C, and melting enthalpy of 287 J/g. Referring to processing of homo-composites, impregnation of UHMWPE fibers by LLDPE matrix could only be attained after melting of LLDPE. However, processing temperature must be lower than the melting point of the UHMWPE fibers, since the fibrous structure must be preserved in order to preserve high mechanical properties. Therefore, the processing window, as reported in Figure 4, was quite narrow, which justifies the choice of the processing temperatures for co-extrusion, compression molding, and the low oven temperature and long processing times used during rotational molding. In fact, low oven temperatures reduce the thermal gradients across the thickness of rotational molded products [17], whereas long processing times allow for efficient sintering of LLDPE powders even at low temperatures [18]. The effect of fiber melting can also be understood by referring to the DSC curve measured during the second scan on UHMWPE, also reported in Figure 4. The most relevant melting properties, i.e., melting peak and enthalpy, were reduced to 139 °C and 168 J/g, respectively. The reduction of the melting peak temperature indicated the reduction of the average lamellar thickness [19], whereas the reduction of the melting enthalpy indicated a reduction of the degree of crystallinity. Both effects were expected to decrease the mechanical properties of the fibers.

DSC analysis was aimed at studying the possible modification of UHMWPE crystalline structure during the processing cycle, including co-extrusion, compression molding, and rotational molding. According to Figure 4, a modification of the crystalline structure of UHMWPE fibers during processing was expected to reduce the melting peak and melting enthalpy measured during the first scan of processed homo-composites. On the other hand, some further stretching of UHMWPE fibers, which could also occur during co-extrusion, was expected to increase the degree of crystallinity and orientation of fibers [20].

The two heating scans performed on the rotomolded homo-composites are reported in Figure 5. In the first scan, two distinct peaks can be observed: The first one, occurred at lower temperatures, and was due to melting of the LLDPE matrix, whereas the second one, occurred at higher temperatures, and was due to melting of the UHMWPE fibers. In the second scan, the two peaks again occurred, but their intensity, area, shape, and position on the temperature axis were significantly modified. As expected, the change in the melting peak between first and second scan was more relevant for UHMWPE compared to LLDPE.

The melting peak temperature of UHMWPE fibers during the first scan was measured to be about 149 °C, which was lower than that of untreated fibers. On the other hand, estimation of the melting enthalpy required a more detailed analysis. In facts, DSC samples included LLDPE matrix and UHMWPE fibers. Despite the fact that the nominal content of fibers was 20% for compression molded samples and 30% for co-extruded samples, the amount of fibers actually present in DSC sample is unknown. This is basically due to the fact that a DSC sample containing such a limited amount cannot be statistically representative of a macroscopic sample. DSC analysis performed on homo-composites provided an apparent melting enthalpy of UHMWPE, which was obtained by dividing the melting area *A_melting,UHMWPE,HC_* to the total sample mass, which, however, comprised both LLDPE and UHMWPE, in unknown amounts:(2)$$\Delta H_{UHMWPE,APP,HC}=\frac{A_{melting,UHMWPE,HC}}{m_{UHMWPE}+m_{LLDPE}}$$

Yet, the true melting enthalpy of UHMWPE was obtained by dividing the melting area to the mass of UHMWPE:(3)$$\Delta H_{UHMWPE,true,HC}=\frac{A_{melting,UHMWPE,HC}}{m_{UHMWPE}}$$

Therefore, by combining Equation (2) and Equation (3), the weight fraction of UHMWPE in DSC samples can be estimated. If the crystalline structure of neat UHMWPE and UHMWPE in the homo-composite is the same, as occurs during the second scan, $\Delta H_{UHMWPE,true,2scan,HC}$ is the same as the melting enthalpy measured on neat fibers, $\Delta H_{UHMWPE,2scan,NF}$. Therefore, the weight fraction of fibers in DSC sample can be estimated as:(4)$$w_{UHMWPE}=\frac{\Delta H_{UHMWPE,APP,2scan,HC}}{\Delta H_{UHMWPE,true,2scan,HC}}=\frac{\Delta H_{UHMWPE,APP,2scan,HC}}{\Delta H_{UHMWPE,2scan,NF}}$$

Once the weight fraction of UHMWPE in DSC sample is known, the value of the true melting enthalpy of UHMWPE measured during the first scan on homo-composite can be estimated as:(5)$$\Delta H_{UHMWPE,true,1scan,HC}=\frac{\Delta H_{UHMWPE,APP,1scan,HC}}{w_{UHMWPE}}$$

However, in this case, due to the different thermal processes of the materials, the melting enthalpy of UHMWPE in the homo-composite can be different from that of neat UHMWPE fibers measured in the first scan. The difference between $\Delta H_{UHMWPE,true,1\;scan,HC}$ in the homo-composite and $\Delta H_{UHMWPE,1\;scan,NF}$ of neat fibers provides an estimate of the reduction of the crystallinity of fibers during the different thermal treatments.

On the other hand, as observed in Figure 4, the DSC analysis of homo-composites showed two distinct peaks, which, particularly in the second scan, partially overlap in a temperature interval around 130 °C. Therefore, in order to calculate the enthalpy associated to each peak, deconvolution of the heat flow signal (HF) was performed by means of sigmoidal curve, known as S-Richards function [21]:(6)$$\begin{array}{c} HF=\Delta H_{1}k_{p1}exp\left(-k_{p1}\left(T-T_{p1}\right)\right){\left[1+\left(d_{p1}-1\right)exp\left(-k_{p1}\left(T-T_{p1}\right)\right)\right]}^{\frac{d_{p1}}{1-d_{p1}}}\\ +\Delta H_{2}k_{p2}exp\left(-k_{p2}\left(T-T_{p2}\right)\right){\left[1+\left(d_{p2}-1\right)exp\left(-k_{p2}\left(T-T_{p2}\right)\right)\right]}^{\frac{d_{p2}}{1-d_{p2}}} \end{array}$$
where ΔH is the apparent melting enthalpy, k_p_ is the intensity factor of the curve (related to its height), d_p_ is the shape factor, which mainly influences the asymmetry of the melting curve, and T_p_ represents the melting peak temperature. The subscripts 1 and 2 refer to the first (LLDPE) and second (UHMWPE) peak. The results of non-linear curve fitting according to Equation (6) are also reported in Figure 5, where the good fit is highlighted by the very high coefficient of determination R^2^ = 0.98. The parameters used for non-linear curve fitting are reported in Table 2. For the sake of clarity, only the parameters relative to the UHMWPE melting peak are reported. Fitting of the UHMWPE melting was obtained, in the second scan, by the use of constant values of the parameters, which effectively indicates that the melting profile was independent on the thermal treatment of the sample, since melting during the first scan deleted all the thermal history of the samples. This observation confirms the hypothesis that during the second scan, crystalline structure of neat UHMWPE and UHMWPE in the homo-composite is the same, which allowed obtaining Equation (4) in the assumption that $\Delta H_{UHMWPE,true,2scan,HC}=\Delta H_{UHMWPE,2scan,NF}$.

The only changing parameter was the peak area factor, $\Delta H_{UHMWPE,APP,\;2scan,HC}$ which however, as previously discussed, is related to the amount of UHMWPE in the sample. Therefore, by the use of Equation (4), an estimation of the amount of UHMWPE was performed: The values obtained and reported in Table 2 are different from the nominal value of 0.2 or 0.3.

In contrast, fitting of the melting peak of UHMWPE during the first scan required the use of different parameters for different samples. In particular, the effect of T_P_ is highlighted: Compared to untreated UHMWPE fibers, processing of the fibers involved a decrease of the melting peak temperature, indicating a reduction of the average crystallite thickness during processing. On the other hand, estimation of $\Delta H_{UHMWPE,true,1\;scan,HC}$ by means of Equation (5) clearly indicates the reduction of the amount of crystalline phase due to fiber processing. The effect was more relevant for compression molded samples, which were processed at 135 °C. Co-extrusion, which was performed by using a die temperature of 130 °C, allowed for a retention of a higher degree of crystallinity in processed samples, with a reduction of the melting enthalpy of about 25% compared to untreated fibers. However, further processing by compression molding and rotational molding involved some reduction of the degree of crystallinity, which, however, was found to be lower than 5%. Both observations, i.e., reduction of melting peak and enthalpy of melting after co-extrusion, highlight that the effect due to partial melting of UHMWPE fibers prevailed over a potential stretching effect.

Besides the changes in the crystalline structure of UHMWPE, a second very relevant issue is related to the efficiency of fiber impregnation during processing. To this purpose, density data, measured by a pycnometer and reported in Table 3, were compared to the theoretical density of the homo-composites, obtained by the rule of mixtures using the density data of the material technical data sheet, which was 0.936 g/cm^3^. For co-extruded homo-composite, the measured density was lower than the theoretical density, which indicated the presence of voids; during co-extrusion of homo-composites, the die pressure build-up (about 10 bar) was not sufficiently high to allow a complete impregnation of the fibers by the molten matrix. Compression molded samples show, in both cases, a density which was comparable to the theoretical density; the reduction of the void fraction x_v_ to values lower than 2% indicated a satisfactory impregnation of the fibers, which was attributed to the much higher pressure (100 bar) attained in the compression molding process, which allowed the molten matrix to flow in-between the UHMWPE fibers.

### 3.2. Mechanical Characterization of Polyethylene Homo-Composites

Typical stress–strain curves from tensile testing of neat fibers, neat matrix, and of polyethylene homo-composites are reported in Figure 6, and the corresponding values measured for the modulus are reported in Table 4.

Neat matrix was characterized by a modulus of about 0.5 GPa, which was equivalent to the value reported in the technical data sheet, whereas UHMWPE fibers were characterized by a modulus of 40 GPa. The modulus of the fibers was measured on the as-received tape, by using the actual cross-section of the tape, estimated as:(7)$$A_{true}=\frac{M}{\rho_{UHMWPE}L}$$
where *M* is the mass of the sample, *L* the length, and *ρ_UHMWPE_* the density of UHMWPE, 0.95 g/cm^3^, according to material TDS.

Polyethylene homo-composites showed a significantly different behavior, based on the processing conditions. Significant reduction of the tensile modulus E compared to the theoretical modulus E_T_, as estimated by the rule of mixtures, can be attributed either to partial fiber melting during processing, as discussed in the analysis of DSC, or to poor fiber impregnation, as discussed previously on density data, resulting in a significant void fraction.

In Table 4, sample HC_CM, processed at 135 °C, showed a relatively low modulus, if compared to the theoretical modulus. However, since in this case the void fraction was very low, the low modulus can be attributed to partial melting of the fibers, occurring due to the relatively high processing temperature, as previously discussed in the analysis of DSC. Sample HC_CE shows a modulus higher than that of HC_CM, but still lower than the theoretical value. In this case, however, besides the effect of partial fiber melting, poor fiber impregnation also occurred. Nevertheless, the higher modulus compared to compression molded samples indicated that the effect of fiber melting was significantly reduced, as also highlighted in the analysis of DSC. Combining co-extrusion and compression molding allowed us to further increase the modulus to a value which was equivalent to the theoretical value.

Plate bending tests were carried out on plates extracted from rotomolded samples. Figure 7 shows the comparison between the unreinforced plate and some of the plates reinforced with the different configurations of the bars, as reported in Table 1. The equivalent bending modulus of the plates, obtained by Equation (1), is reported in Figure 8. The modulus of neat LLDPE produced by rotational molding was noticeably the same as that of LLDPE processed by compression molding, indicating the efficiency of sintering during processing, which allowed us to obtain void-free products even when relatively low oven temperatures were used. All the homo-composite prototypes showed a significant increase in the modulus, with the highest value reached when a 3bar_HC_CM configuration was used. Results reported in Figure 9 show that the bending modulus mainly depended on the amount of fibers, and that the effect of fiber layout (crossing or parallel bars) was quite marginal.

Figure 10 shows the results of impact tests on unreinforced plate and sample 8bar_HC_CE_CM, extracted from rotational molded prototypes. The unreinforced plate was able to withstand an energy of 25 J, but upon impact by a 30 J energy, the sample underwent failure, as observed by the interruption of the energy curve during the dart rebound phase. However, the homo-composite 8bar_HC_CE_CM was able to withstand an energy up to 40 J. 

The results reported in Figure 7, Figure 8 and Figure 9 show that, due to the addition of only 4% of UHMWPE fibers, the stiffness of polyethylene homo-composite became three times higher than that of neat LLDPE. In Figure 10, in view of the fact that the homo-composite reinforced with eight bars did not break even for an impact energy of 40 J, whereas the neat LLDPE failed for an energy of 30 J, a toughness increase of at least 33% can be estimated. In particular, a reduction in the toughness was the main issue associated to the use of more conventional strategies for reinforcing of LLDPE processed by rotational molding, as found by different authors. 

The addition of fillers can be considered as a first option in order to improve the stiffness of rotational molded LLDPE. For example, in reference [4], the addition of spherical glass beads involved an increase of the modulus of LLDPE by about 45%, with a decrease of the strength by 200%. Similar results were also found in [22], where the addition of natural fillers involved a modulus increase from 300 to 500 MPa, but a strength reduction from 18 to 5 MPa.

Alternatively, longer fibers can be used for a more efficient stiffening of LLDPE. The incorporation of agave fibers increases the modulus of LLDPE by about 60%, with a decrease of the impact strength by 400% [23]. Maple fibers have a similar stiffening effect on LLDPE, but involve an even more significant embrittlement, as found in [24], where the strain at break was found to decrease from 1300% to 50% upon the addition of 10% of maple fibers.

Finally, the addition of nano-fillers, such as halloysite, has been shown to have a similar effect on the mechanical properties of LLDPE, as reported in [25], where an increase of the modulus from 700 to 850 MPa was brought by a toughness decrease from 200 to 50 J/m.

In general, all the cited works indicate that the decrease of the ductility and/or toughness of LLDPE, is much more significant than the modulus increase.

However, the potential advantages of the use of UHMWPE fibers, rather than pultruded glass fibers used in a previous work [13], can be highlighted. In fact, in the latter case, the maximum stiffness attained was 45% higher than that of the neat LLDPE, despite the fact that glass fibers are characterized by a tensile modulus of 70 GPa, whereas UHMWPE fibers have a lower modulus, around 40 GPa, as reported in Table 4. However, owing to the very similar coefficients of thermal expansion of UHMWPE and LLDPE, up to 4% of the fibers was added. In contrast, processing by rotational molding does not allow for an increase in the amount of pultruded glass fibers to values higher than 0.6%; higher values resulted in significant distortion of the prototypes, due to a significant difference in thermal expansion coefficients between LLDPE and glass.

## 4. Conclusions

In this work, an innovative methodology was developed for the production of polyethylene homo-composites to be processed by rotational molding. In contrast to conventional reinforcing, in which fillers or fibers are randomly and uniformly dispersed in the matrix, the proposed approach is based on the selective reinforcement of low modulus LLDPE by the use of UHMWPE fibers.

The process for inclusion of UHMWPE fibers in LLDPE matrix and processing by rotational molding required the preliminary production of reinforcing homo-composite bars by co-extrusion or compression molding. The production of the reinforcing bars was required in order to allow impregnation of the fibers by the molten matrix, which cannot occur during the pressure-free rotational molding cycle.

Therefore, a preliminary evaluation of different processing routes for the production of reinforcing bars made of UHMWPE fibers and LLDPE matrix was performed. The different processes considered include compression molding, which is characterized by a very simple and fast processing, co-extrusion, which requires proper design and modification of extrusion die, and a combination of the two processes. The main advantage of compression molding is the very efficient impregnation of the fibers, which, however, can be attained at relatively high temperatures, which involves the partial melting of the fibers and a significant loss of the mechanical properties. On the other hand, co-extrusion, which can be performed at lower temperatures, allows for a better retention of the crystalline structure of the fibers, but does not allow for an efficient impregnation of the fibers. A combination of the two processes was found to provide the best mechanical properties, since it allows for the efficient impregnation of the fibers with a reduced melting of UHMWPE. 

In the second step, the reinforcing bars were placed on the inner wall of a hollow mold for subsequent rotational molding. Melting of LLDPE during a standard rotational molding cycle allowed for fiber inclusion into the rotational molded prototype, and therefore for the production of a hollow polyethylene homo-composite component.

Using UHMWPE as reinforcement for LLDPE allowed for an increase in the amount of fibers up to 4%, compared to 0.6%, which was the maximum value attainable with glass fibers. No distortion due to thermal stresses was observed for the cubic prototype. Due to the higher amount of fibers, polyethylene homo-composite produced by rotational molding showed a stiffness which was about times that of unreinforced LLDPE, and about two times that of LLDPE reinforced by pultruded glass fibers, with a negligible increase of the weight of the component. In addition, the produced homo-composite prototypes showed a better impact resistance compared to neat matrix, which has never been attained by the use of other stiffening strategies, which, instead, usually involve a significant embrittlement of LLDPE.

## Figures and Tables

**Figure 1 polymers-11-00528-f001:**
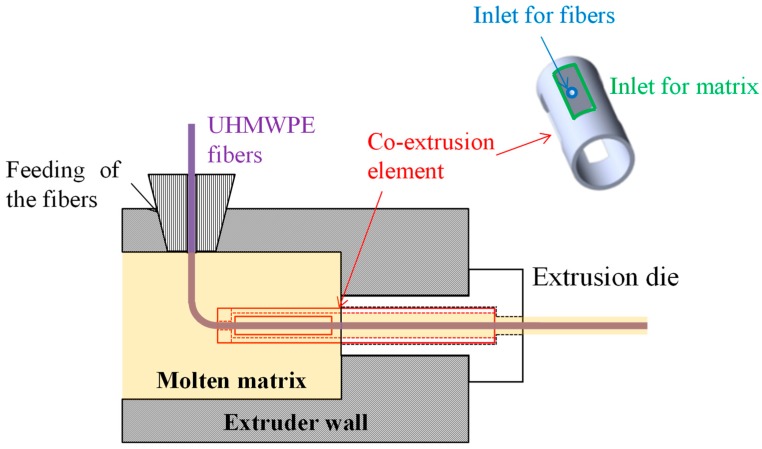
Scheme of co-extrusion.

**Figure 2 polymers-11-00528-f002:**
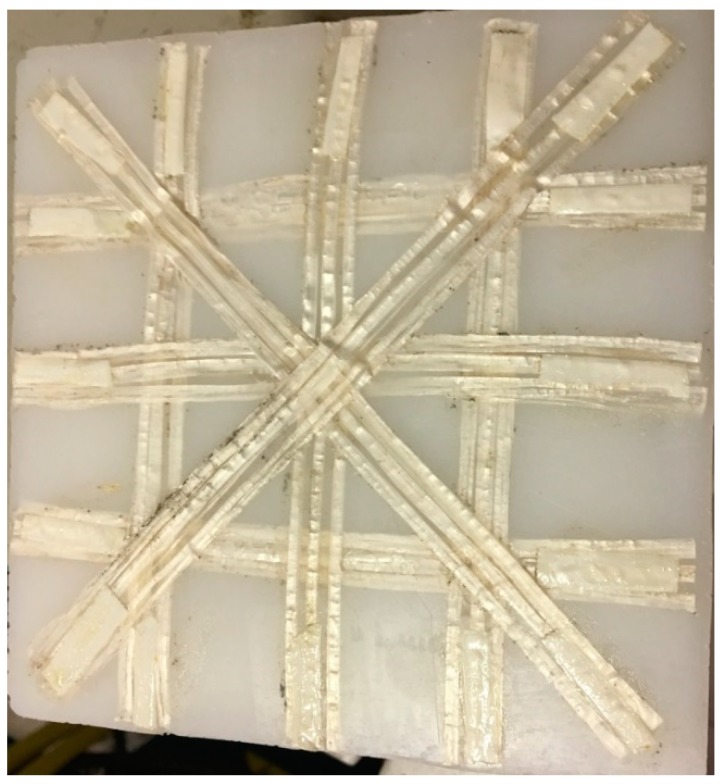
Picture of rotational molded homo-composite prototype.

**Figure 3 polymers-11-00528-f003:**
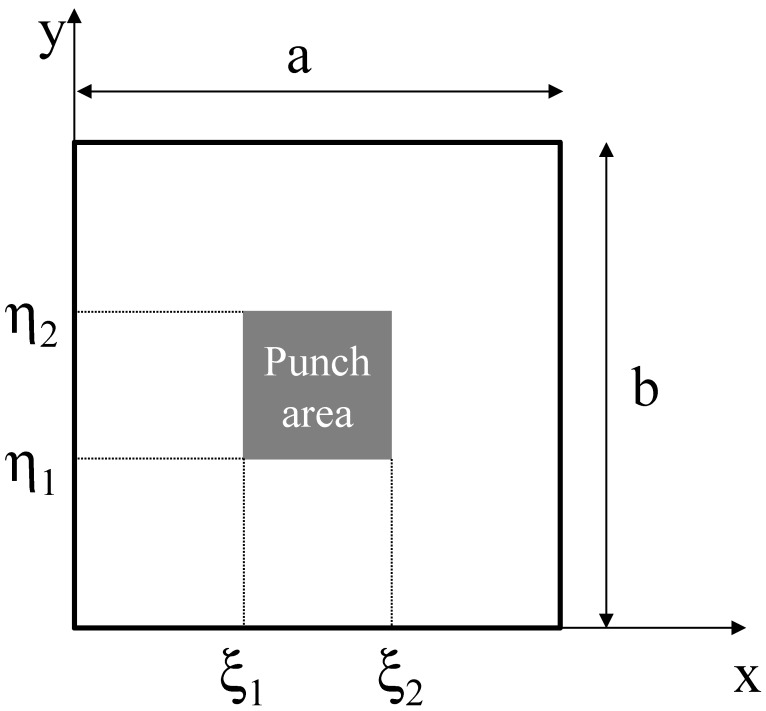
Scheme of the sample geometry and loading device for plate bending tests.

**Figure 4 polymers-11-00528-f004:**
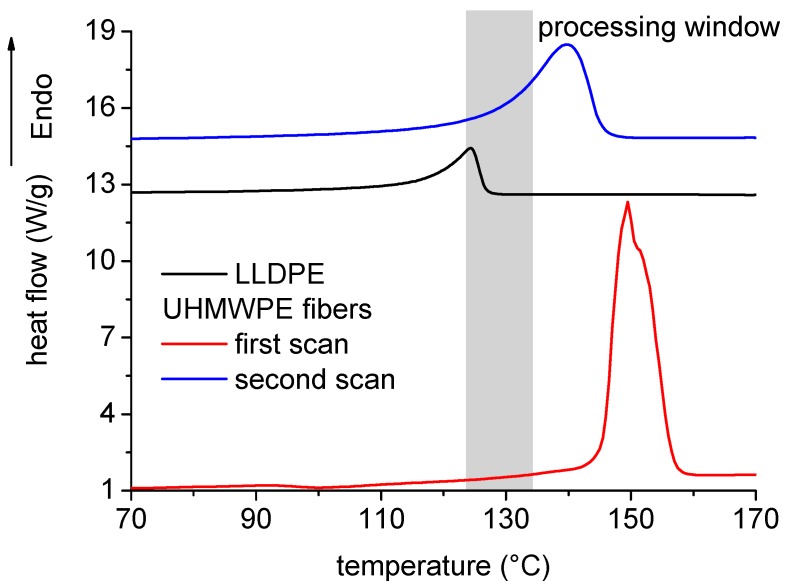
DSC curves on ultra-high molecular weight polyethylene (UHMWPE) fibers and linear low-density polyethylene (LLDPE) matrix.

**Figure 5 polymers-11-00528-f005:**
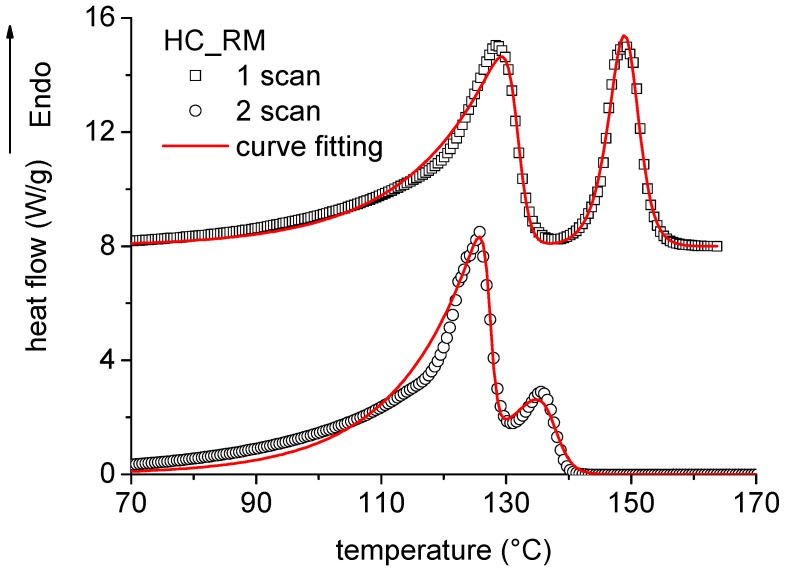
DSC curves on rotational molded samples and nonlinear curve fitting according to Equation (6).

**Figure 6 polymers-11-00528-f006:**
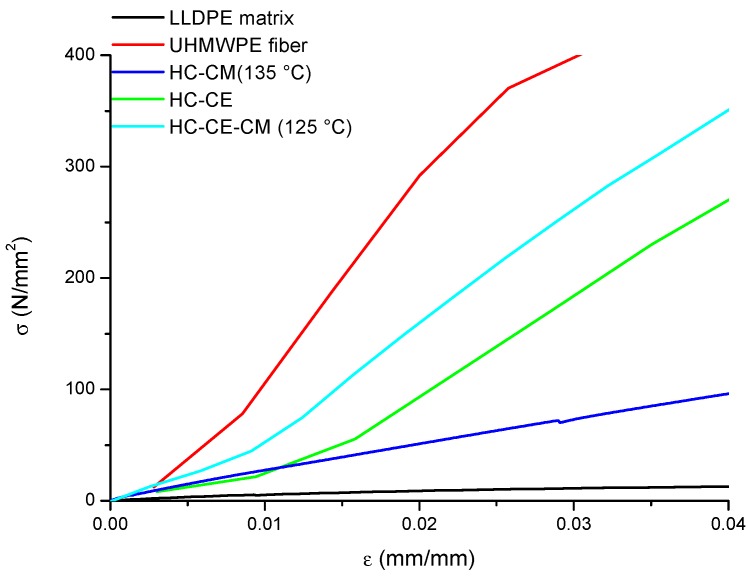
Stress–strain curves on LLDPE matrix, UHMWPE fibers, and homo-composites.

**Figure 7 polymers-11-00528-f007:**
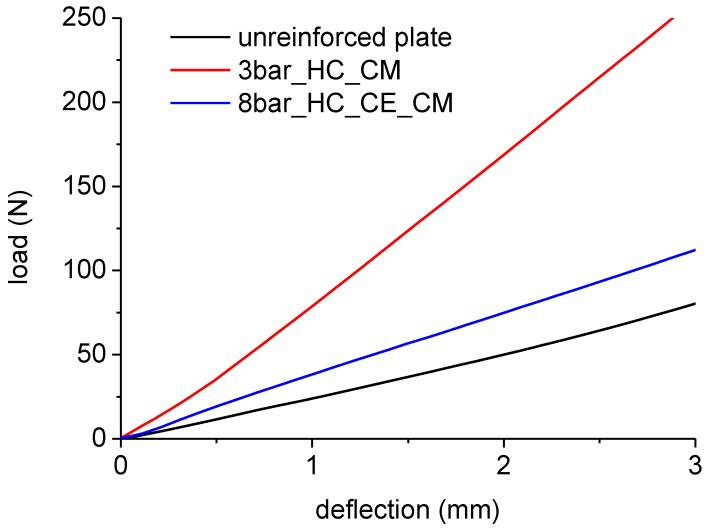
Force–deflection curves from plate bending tests on samples extracted from rotomolded prototypes.

**Figure 8 polymers-11-00528-f008:**
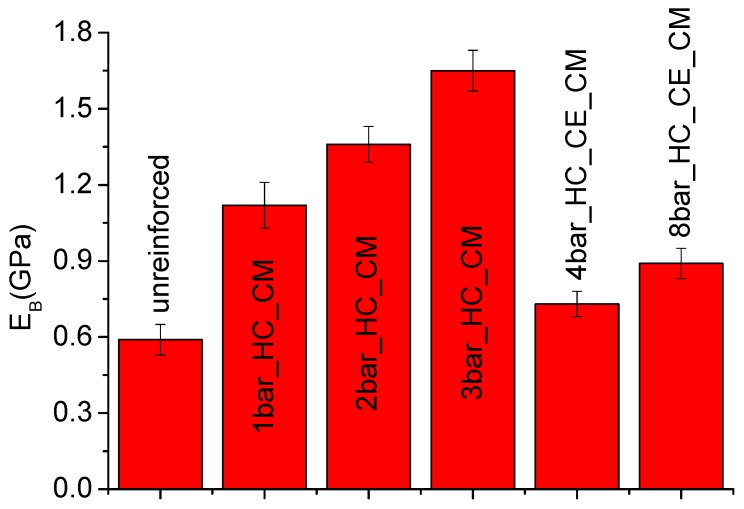
Equivalent bending modulus of unreinforced and reinforced plates.

**Figure 9 polymers-11-00528-f009:**
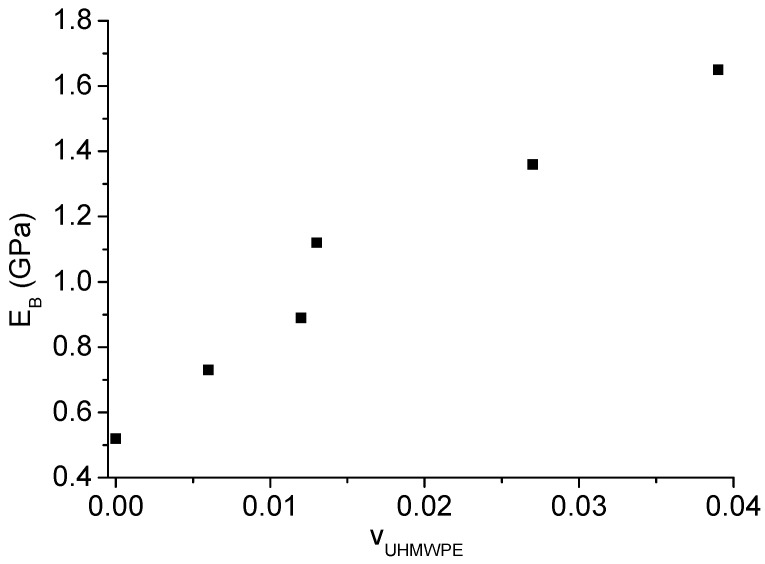
Equivalent bending modulus of reinforced plates as a function of UHMWPE fiber volume fraction.

**Figure 10 polymers-11-00528-f010:**
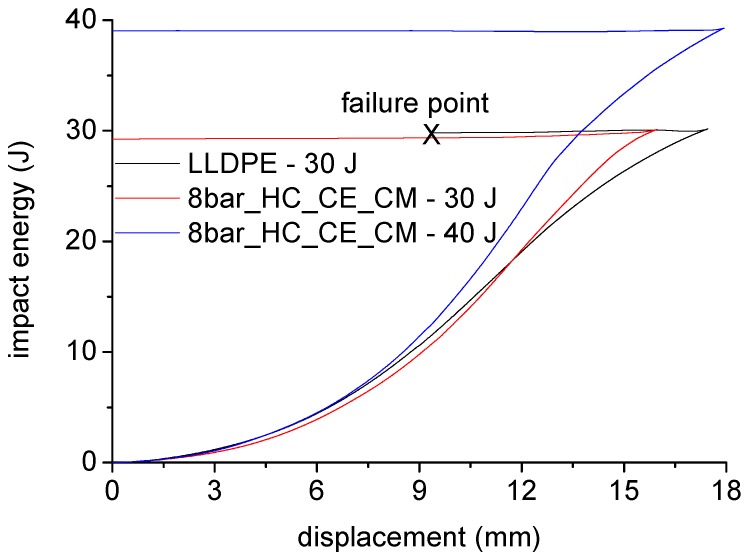
Results of impact tests on reinforced and unreinforced plates.

**Table 1 polymers-11-00528-t001:** Reinforcing bar layout in rotational molded prototypes.

Sample Code	Reinforcing Homo-Composite	Reinforcement Layout	% of UHWMPE Fibers
1bar_HC_CM	Compression molded	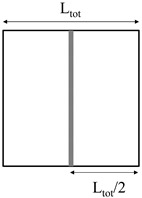	1.3
2bar_HC_CM	Compression molded	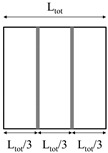	2.7
3bar_HC_CM	Compression molded	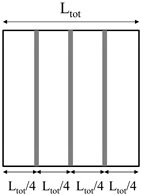	4
4bar_HC_CE_CM	Co-extruded and compression molded	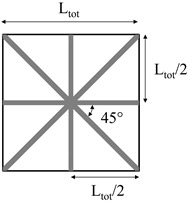	0.6%
8bar_HC_CE_CM	Co-extruded and compression molded	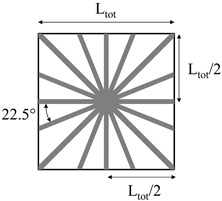	1.2%

**Table 2 polymers-11-00528-t002:** Thermal properties of polyethylene homo-composites.

**2 Scan**	$\Delta H_{UHMWPE,APP}(J/g)$	**d_p_**	**K_p_ (K^−1^)**	**T_p_ (°C)**	$w_{UHMWPE}$ **Equation (4)**	$\Delta H_{UHMWPE,true,1\;scan,HC}$ **Equation (4) (J/g)**
Untreated	158	7.8	0.737	137		
HC_CM	36	7.8	0.737	137	0.23	
HC_CE	42.3	7.8	0.737	137	0.27	
HC_CE_CM	50.7	7.8	0.737	137	0.32	
HC_RM	37.9	7.8	0.737	137	0.24	
**1 Scan**	$\Delta H_{UHMWPE,APP}$ **(J/g)**	**d_p_**	**K_p_ (K^−1^)**	**T_p_ (°C)**		
Untreated	288	1.7	0.31	152		
HC_CM	42	2.0	0.49	148		183
HC_CE	56.7	1.7	0.46	150		212
HC_CE_CM	65.3	1.4	0.44	150		204
HC_RM	48.7	2.5	0.70	149		203

**Table 3 polymers-11-00528-t003:** Density of polyethylene homo-composites after different processing.

Sample	$\rho_{s}(g/{\mathrm{cm}}^{3})$	x_v_
HC_CM	0.93 ± 0.04	0.006
HC_CE	0.82 ± 0.03	0.12
HC_CE_CM	0.92 ± 0.04	0.017

**Table 4 polymers-11-00528-t004:** Tensile modulus of polyethylene homo-composites.

Sample	*E* (GPa)	*E_T_* (GPa)
LLDPE matrix	0.6	-
UHMWPE fibers	40	-
HC_CM (135 °C)	2.37	8.40
HC_CE	9.07	12.35
HC_CE_CM (125 °C)	11.10	12.35

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
