# Peer review of "Thermal and Mechanical Analysis of Polyethylene Homo-Composites Processed by Rotational Molding"

_polymers, 2019, doi:10.3390/polym11030528_

Round 1

Reviewer 1 Report

The paper evaluates the effect of UHMWPE fibers on the thermal and mechanical properties of roto-molded LLDPE. Different processes were used to produce the homo-composites, which were then roto-molded. Results show improved mechanical properties and low density increase.

1. The title should be more specific to the characterization of the properties of the homo-composites. The main focus of the results discussed in the submission is the compounding of the UHMWPE with the LLDPE. Processing is not discuss in depth, so I will suggest changing the title to something more specific. 

2. The `Materials and Methods` and the `Experimental results` section are very wide. I would suggest using some subsections to improve clarity for the reader.

3. English language should be improved. Some sentences are not properly written from a grammatical perspective. This makes interpretation somehow confusing.   

4. line 137, provide a reference for the model used for eq.1

5. lines 243-245. `...seems to be quite marginal.` I would recommend the use of more specific and quantitative sentences to discuss findings in the paper. 

6. Table II. The alignment of the columns should be fixed to improve reading quality.

7. line260 `Figure 5Table`. This looks like a typo.

8. line 273. Where and why are these voids formed? How was the density measured?

9. Figure 5 and other similar plots. A different style of plotting should be selected, some lines cannot be distinguished.

10. lines 307-333. The comparison with other reinforcing solutions should be supported by experimental evidence. As it stands, it does not add any scientific value to the manuscript. 

11. line 345. ` ...a new design and processing route...`: the innovation in the design and processing presented in the manuscript should be better highlighted. What is the novelty introduced in the rotomolding processing? 

12. line 350 and line 89. How was the die modified to allow co-extrusion? This should be discussed.

13. More quantitative results should be discussed in the Conclusions.

14. line 359. How was quality of the rotomolded parts checked? Planar faces are mentioned here for the first time. 

For these reasons, to me the paper should be rejected because it

does not suit the quality

9

standards of the journal.

For these reasons, to me the paper should be rejected because it

does not suit the quality

9

standards of the journal.

For these reasons, I recommend the revision of the paper before publication.

Author Response

see the attached score sheet for a point by point response to the reviewer's comments

Reviewer 2 Report

In this work, a design and processing route is developed for the production of polyethylene homo-composite to be processed by rotational molding. A preliminary evaluation of different processing routes for the production of reinforcing bars made of UHMWPE fibers and LLDPE matrix is performed.

This work can be considered for publication after revisions as follows:

1)      Some quantitative statements in the abstract and conclusions are essential for an original scientific paper. I will recommend to revise abstract and conclusions accordingly.

2)      “Experimental Results” should be replaced by “Results and discussion” (line 149)

3)      There are few grammatical errors in the manuscript. Manuscript should be edited for English.

4)      Line 41: Reference formatting need to be changed (throughout the article)

5)      Line 137: Need a reference for the equation.

6)      Figure 3: correct the spelling on Y-axis. Need to mention Exo and Endo on the plot.

7)      Figure 5: There is a mistake in the unit of X-axis. It is mm/min.  

Author Response

(The authors gave the same response as above.)

Reviewer 3 Report

Materials and methods 

Page 2 line 85 second method. It is mentioned co-extrusion. This must be modified as the equipment is twin-screw extrusion, so not co-extrusion methodology. Also, the explanation of non-melting of UHMWPE at 130°C must be explained in a proper manner. SO the materials and methods of blends is unclear.

Related to crystallinity of UHMWPE (page 7) it is advised to integrate the explanation of possible stretching effects during twin screw blending/processing, as internal stresses can be initiated and affect the crystallinity of the blend, specially for UHMWPE.

The methodology of reinforcements within the final rotomolded part must be explained in a more clear manner. Now, the goal and reason of reinforcements within final rotomolded parts is unclear.

It is noticed that references are mainly related to the first author. It is absolutely requested also to refer to other research works in the scope of rotomolding and material characterization, specially within the state of the art of rotomoulding materials, processing and related characterization. this will strengthen the research work and explain comparison with other state of the art results.

Author Response

(The authors gave the same response as above.)

Round 2

Reviewer 3 Report

The paper has been well improved after revision.